# Screening for Glucose Metabolism Disorders, Assessment the Disse Insulin Resistance Index and Hospital Prognosis of Coronary Artery Bypass Surgery

**DOI:** 10.3390/jpm11080802

**Published:** 2021-08-17

**Authors:** Alexey N. Sumin, Natalia A. Bezdenezhnykh, Andrey V. Bezdenezhnykh, Anastasia V. Osokina, Anastasiya A. Kuz’mina, Anna V. Tsepokina, Olga L. Barbarash

**Affiliations:** Federal State Budgetary Institution, Research Institute for Complex Issues of Cardiovascular Diseases, Sosnovy Blvd. 6, 650002 Kemerovo, Russia; n_bez@mail.ru (N.A.B.); andrew22014@mail.ru (A.V.B.); av.osokina80@yandex.ru (A.V.O.); stusha76@mail.ru (A.A.K.); annacepokina@mail.ru (A.V.T.); olb61@mail.ru (O.L.B.)

**Keywords:** coronary bypass surgery, carbohydrate metabolism disorders, postoperative complications, insulin resistance markers

## Abstract

Objective: To study insulin resistance markers and their relationship with preoperative status and hospital complications of coronary artery bypass grafting (CABG) in patients with type 2 diabetes, prediabetes and normoglycemia. Methods: We included 383 consecutive patients who underwent CABG. Patients were divided into two groups—with carbohydrate metabolism disorders (CMD, *n* = 192) and without CMD (*n* = 191). Free fatty acids and fasting insulin in plasma were determined, and the Disse, QUICKI and revised QUICKI indices were calculated in all patients. Perioperative characteristics and postoperative complications were analyzed in these groups, and their relations with markers of insulin resistance. Results: Screening before CABG increased the number of patients with CMD from 25.3% to 50.1%. Incidence of postoperative stroke (*p* = 0.044), and hospital stay after CABG > 30 days (*p* = 0.014) was greater in CMD patients. Logistic regression analysis revealed that an increase in left atrial size, age, aortic clamping time, and decrease in Disse index were independently associated with hospital stay >10 days and/or perioperative complications. Conclusions: Screening for CMD before CABG increased the patient number with prediabetes and type 2 diabetes. In the CMD group, there were more frequent hospital complications. The Disse index was an independent predictor of long hospital stay and/or poor outcomes.

## 1. Introduction

The high prevalence of diabetes mellitus (DM) is a recent worldwide trend [1]. The frequent comorbidity of diabetes mellitus and atherosclerosis of various localization ultimately leads to the need for myocardial revascularization [2]. The FREEDOM study showed that CABG is preferable to PCI in diabetic patients with multivessel coronary disease, reducing mortality and the rate of myocardial infarctions [3], which was confirmed by prospective observation [4] and in subsequent meta-analysis [5]. Accordingly, a trend of an increased number of performed CABG compared to PCI was noted in some centers [6], as well as increased attempts to identify previously undiagnosed diabetes mellitus or prediabetes before CABG surgery. Several studies have shown that disorders of carbohydrate metabolism negatively affected the results of myocardial revascularization [7,8,9,10]. However, some data could not support these statements, in particular, Djupsjo C. et al. did not reveal an effect of newly diagnosed diabetes and previously known pre-diabetes on the long-term survival in patients who underwent CABG [11]. Additionally, not all studies noted an increase in the number of CABGs performed in multivessel coronary artery disease in diabetes mellitus [12]. Therefore, the search for optimal treatment tactics in this category of patients continues [13,14,15]. Attempts are made to use markers of diabetes compensation (glycated hemoglobin, fructosamine) for this purpose; data possibly supporting a relationship of these indicators with the outcomes of CABG were obtained [16]. In this regard, attention is drawn to studying insulin resistance, which can contribute to changes in the function of blood vessels and the heart even with normoglycemia [17], and, potentially, can affect the outcomes of coronary bypass grafting, leveling the effect of carbohydrate metabolism disorders on them. Currently, the association of insulin resistance with coronary artery disease severity is being actively studied; however, studies of insulin resistance in patients with heart surgery are sparse [18]. Researchers have proposed several markers of insulin resistance, so the question of their diagnostic significance remains relevant [19]. This served as the basis for the present study, the purpose of which was to study various markers of insulin resistance and their relationship with preoperative status and in-hospital complications of coronary artery bypass grafting (CABG) in patients with disorders of carbohydrate metabolism (prediabetes and type 2 diabetes mellitus) and normoglycemia.

## 2. Subjects, Material and Methods

This single-center, observational study was conducted at the Research Institute for Complex Issues of Cardiovascular Diseases, Kemerovo. Consecutive patients who underwent elective CABG in the cardiovascular surgery department of the clinic from 1 October 2011 to 22 March 2012 were included. The study design is shown in Figure 1. In total, the study involved 396 consecutive patients who were planned for CABG; in 6 of them, the revascularisation tactics were revised to percutaneous intervention due to the comorbidity, 7 patients were denied myocardial revascularization. Thus, CABG was performed in 383 patients included in the study. Upon admission to the hospital for preparation for CABG, in all patients glycemic status was examined, free fatty acids (FFA) and fasting insulin were determined, and the Disse insulin resistance index, QUICKI (Quantitative Insulin Sensitivity Check Index) and Revised-QUICKI were calculated.

### 2.1. Data Collection

Demographic and perioperative patient data were obtained from the electronic database of the institute’s CABG registry (certificate No. 2012020868 on registration of CABG database). For each patient who met the inclusion criteria, the following data were collected: age, sex, body mass index, present or absent history of myocardial infarction, hypertension, stroke, diabetes mellitus, heart failure, PCI, CABG, peripheral artery stenosis, dyslipidemia, smoking, creatinine and glucose levels. In addition, intraoperative parameters were recorded, including use of cardiopulmonary bypass; duration of surgery; aortic clamping time; number of grafts; and combined surgeries (ventriculoplasty, thrombectomy, radiofrequency ablation, carotid endarterectomy). The data of anamnesis, laboratory examinations, echocardiography (extended protocol), coronary angiography, ultrasound and angiographic examination of the aorta, brachiocephalic and peripheral arterial basins, characteristics of the operation, duration of hospitalization, frequency of postoperative complications in groups were analyzed. Confirmation of the presence and assessment of the prevalence of atherosclerotic lesions were carried out using color duplex scanning of the extracranial arteries and the lower extremities arteries (apparatus “Aloka 5500”).

### 2.2. Determination of Glycemic Status

In patients with borderline fasting hyperglycemia (6.1–6.9 mmol/L (110–125 mg/dL)) and no previously established diabetes mellitus, as well as in patients with previously known prediabetes, an oral glucose tolerance test (OGTT) was performed, unless contraindicated. If the results of several studies of fasting, postprandial glycemia or glycated hemoglobin were sufficient to establish a diagnosis of diabetes, OGTT was not performed. The diagnosis of type 2 diabetes mellitus and other carbohydrate metabolism disorders (CMD) was established by an endocrinologist in accordance with the current criteria for modern classification of diabetes mellitus and other glycemic disorders [20]. In the case of a newly established carbohydrate metabolism disorder, the following criteria were used for diagnosis. In the absence of previously established diabetes mellitus and in the presence of borderline fasting hyperglycemia (6.1–6.9 mmol/L) or previously known prediabetes, an oral glucose tolerance test (OGTT) was performed and glycated hemoglobin was determined. Diabetes mellitus was established in the case of fasting glycemia ≥ 7.0 mmol/L, glycemia at 120 min of OGTT ≥ 11.1 mmol/L; blood glucose at random determination ≥ 11.1 mmol/L in the presence of typical symptoms of hyperglycemia (polydipsia, polyuria, weakness). In the absence of symptoms of acute metabolic decompensation, the diagnosis of diabetes mellitus was established on the basis of two digits in the diabetic range, for example, double-determined blood glucose or a single determination of HbA1c + single determination of blood glucose. The level of glycated hemoglobin HbA1c ≥ 6.5% corresponded to diabetes mellitus [20]. When determining prediabetes (impaired fasting glycemia, impaired glucose tolerance) we also used the WHO 1999–2013 criteria, which are adopted in our country, and not the more stringent criteria of the American Diabetes Association. According to WHO criteria Impaired glucose tolerance (IGT) was diagnosed with fasting plasma glucose < 7.0 mmol/L (126 mg/dL) and 2 h glucose 7.8–11.1 mmol/L (140–200 mg/dl). Impaired fasting glucose (IFG) was diagnosed with fasting plasma glucose between 6.1 and 6.9 mmol/L (110–125 mg/dL) and (if measured) plasma glucose after 2 h <7.8 mmol/L (<140 mg/dl). HbA1c level up to 6.0% was considered normal, HbA1c level 6.0–6.4% corresponded to prediabetes. The term prediabetes was understood as impaired fasting glucose (IFG) or impaired glucose tolerance (IGT), or a combination of both.

The described sample (*n* = 383) included no patients with type 1 diabetes mellitus and other types of diabetes mellitus not related to type 2, therefore, when the term “diabetes mellitus” is mentioned in the text of this article, it means “type 2 diabetes mellitus”, unless otherwise specified. Screening before coronary artery bypass grafting increased the number of patients with established diabetes mellitus from 20.4% (*n* = 78) to 32.6% (*n* = 125), the number of persons with prediabetes from 5.0% (*n* = 19) to 17.5% (*n* = 67), the total number of persons with any established disorders of carbohydrate metabolism from 25.3% (*n* = 97) to 50.1% (*n* = 192) (Figure 1). More than a third of all diabetes mellitus (37.6%) and the majority of prediabetes cases (71.6%) were detected during additional preoperative examination. Patients with normoglycemia accounted for 49.9% of all (*n* = 191). For further analysis, the following sample was formed: 383 patients were divided into 2 groups according to their glycemic status: Group 1 contained patients with CMD (type 2 diabetes mellitus or prediabetes (IFG, IGT), *n* = 191, group 2 was formed by patients without established carbohydrate disorders exchange (*n* = 192) (Figure 1).

### 2.3. Hospital Postoperative Complications

CABG complications included the following: intra- and postoperative myocardial infarction (MI), defined by the presence of a “new” Q wave on the ECG, changes in the ST-T segment, accompanied by a decrease in the left ventricular ejection fraction and/or an increase in troponin I; heart failure requiring long-term inotropic support; paroxysms of atrial fibrillation; stroke; acute kidney injury; progression of renal failure in patients with chronic kidney disease (CKD), renal replacement therapy. The concentration of serum creatinine was determined before surgery one or more times according to indications. In the case of repeated measurements, the results of the determination closest to the date of the CABG were taken into account. In accordance with the diagnostic criteria of the Acute Kidney Injury Network (AKIN), acute kidney injury after surgery was diagnosed in the case of an increase in creatinine 1.5 times or more in comparison with the baseline value, or an increase of more than 26.4 μmol/L; or urine output <0.5 mL/kg per hour for 6 h or more [21]. After surgery, serum creatinine values were recorded daily, diuresis-hourly in the intensive care unit. Other hospital complications that were taken into account were multiple organ failure; pneumonia, respiratory failure, significant complications from the sternal wound: purulent complications (with a severe inflammatory reaction, diastasis of the wound edges, healing by secondary intention), sternum diastasis, mediastinitis, bleeding, remediastinotomy for bleeding or mediastinitis. The hospital mortality (all deaths after CABG during the hospital stay) rate was analyzed. A combined endpoint was used to analyze hospital outcomes. The combined endpoint was a hospital stay > 10 days or any of the postoperative complications described in this section, including fatal complications.

#### Perioperative Glycemic Management

Diabetic patients underwent daily monitoring of glycemia, followed by an examination by an endocrinologist and selection of antihyperglycemic therapy. Glycated hemoglobin (HbA1c) in all patients with established diabetes before CABG was determined by turbidimetric inhibitory immunoassay of hemolyzed whole blood. The method for the determination of HbA1c is certified in accordance with the National Glycohemoglobin Standardization Program (NGSP) and standardized for Diabetes Control and Complications Trial (DCCT) studies. Preoperative preparation of patients with diabetes included achieving target levels of carbohydrate metabolism under an endocrinologist’s control, discontinuation of oral antihyperglycemic drugs and prescribing insulin according to indications (base-bolus regimen or short-acting insulin). Perioperatively, all patients underwent glycemic control in the intensive care unit with relief of hyperglycemia with short-acting insulin (intravenous, subcutaneous), followed by monitoring the therapy effectiveness. Perioperative glycemic management was carried out in accordance with the current national recommendations at that time [22].

### 2.4. Evaluation of Indicators of Lipid Metabolism and Insulin Resistance Indices

Lipid spectrum indices (cholesterol, triglycerides (TG), high-density lipoprotein (HDL) cholesterol, were determined by enzymatic colorimetric methods (sets of reagents Cholesterol FS “DDS”, “Triglycerides FS” DDS “and” HDL cholesterol “JSC” Diacon DS “). The values of LDL cholesterol and atherogenic index were determined by calculation methods. Evaluation of lipid spectrum parameters was carried out by a fasting blood test in all patients upon admission to the hospital to prepare for CABG. To determine free fatty acids, Thermo Fisher Scientific reagents (Erlangen, Germany) were used. Insulin levels were determined using Accu-Bind ELISA Microwells test systems from Monobind Inc BCM Diagnostics (Lake Forest, CA, USA). Free fatty acids (FFA) and fasting insulin were determined in 383 consecutive patients of this sample, in the same patients the Disse insulin resistance index, QUICKI (Quantitative Insulin Sensitivity Check Index) and Revised-QUICKI were calculated. 

The Disse index was calculated from the Disse equation: DI = 12 × (2.5 × {[serum HDL cholesterol/total cholesterol (mmol/L)] − [serum FFA (mmol/L)]}) − fasting serum insulin (IU/mL). As the Disse index value is always calculated below zero, an increase in the value corresponds to a decrease in IR [23]. QUICKI was calculated using the formula QUICKI = 1/[log (I0) + log (G0), where I0 is basal glycemia (mg/dL), G0 is basal insulinemia (mIU/mL). Revised-QUICKI was calculated using the formula Revised-QUICKI = 1/(log (glucose) + log (insulin) + log (FFA)). The decrease in the QUICKI and Revised-QUICKI indices corresponded to the decrease in insulin resistance.

### 2.5. Statistical Analyses

Statistical processing was carried out using the standard STATISTICA 8.0 software package.

Quantitative data distribution was checked using the Shapiro–Wilk test. Due to the fact that the distribution of all quantitative traits differed from normal, they were described using the median indicating the upper and lower quartiles (25th and 75th percentiles). For comparison of groups, the Mann–Whitney test and χ^2^ (chi-square) were used. With a small number of observations, Fisher’s exact test was used with Yates’ correction. Stepwise multiple linear regression analysis was used to assess the relationship between FFA and Disse’s index with perioperative parameters. Binary logistic regression (Forward likelihood ratio) was used to identify predictors of long hospital stay or poor outcomes. The level of critical significance (*p*) during the regression analysis was taken equal to 0.05.

## 3. Results

The main characteristics of the patients are presented in Table 1. Patients of the two groups did not differ in terms of the median age; there were significantly fewer men and smokers in the prediabetes and DM 2 groups (Table 1). 

Higher median body mass index, greater prevalence of obesity and arterial hypertension was observed among the patients with CMD, compared to normoglycemic persons. Patients of the two groups had no difference in the cardiovascular events incidence and vascular interventions history, or in the preoperative risk assessment results. The groups were comparable in terms of the rate of simultaneous operations, surgery duration, bypass and aortic clamping duration and intraoperative blood loss (Table 1). Patients with CMD stayed in the hospital longer after CABG compared to those without CMD: the differences were significant both in the median days of hospital stay (*p* = 0.015) and in the proportion of people staying for longer than 10 days (*p* = 0.005) and longer than 30 days (*p* = 0.024). Regarding preoperative drug management, beta-blockers and calcium antagonists were prescribed in the CMD group more often, while the rest of the main drug therapy had no difference between the groups (Table 1). Only patients with type 2 diabetes received antihyperglycemic drugs (Table 1). Oral antihyperglycemic drugs were discontinued before CABG and insulin was given as needed. The median HDL cholesterol was lower (*p* = 0.004), and the median of triglycerides was significantly higher (*p* < 0.001) in the CMD group than in the normoglycemic group (Table 2). The patients of the two groups did not differ in terms of renal filtration and coagulation tests results, with the exception of SFMS, the median of which was higher in the CMD group (*p* = 0.014).

The median fasting glucose and glycated hemoglobin were consistently higher in the CMD group (*p* < 0.001). The median of free fatty acids (FFA) in this group was also significantly higher (*p* < 0.001). At the same time, the medians of insulin levels and the calculated indices of insulin resistance (Disse, QUICKI, and Revised-QUICKI) were comparable. 

According to the results of echocardiography left ventricle (LV) myocardium mass, LV end-systolic and end-diastolic volumes, as well as LV end-systolic dimension, were higher in patients with CMD than in the normoglycemic group (*p* = 0.009, *p* = 0.042 and *p* = 0.006, respectively), LV ejection fraction was significantly lower in the CMD group (*p* = 0.037). In addition, the median sizes of the left atrium were higher in patients with CMD (*p* = 0.009). The rest of the echocardiographic parameters were comparable. There were no differences between groups in the number of affected main coronary arteries and in the incidence of non-coronary stenosis (Table 2).

Multiple regression was run to predict FAA from age, body mass index, EuroSCORE II, aortic clamping time, bypass duration, APTT, SFMS, fibrinogen, glucose, triglycerides, LV myocardial mass index, length of stay, heart rate, left atrium. Triglycerides, glucose, heart rate, body mass index and APTT statistically significantly predicted FAA level. F (2, 191) = 13.397; *p* < 0.0001; R^2^ = 0.265 (Table 3, Appendix A). 

When predicting the relationship of factors with the Disse index, age, body mass index, EuroSCORE II, heart rate, aortic clamping time, bypass duration, length of stay, echocardiographic data, free fatty acids, triglycerides, glucose, APTT, fibrinogen, SFMS were included in the multiple regression model. Only interventricular septum thickness and LV ejection fraction statistically significantly predicted Disse index: F (2, 108) = 5.222, *p* = 0.009, R^2^ = 0.179 (Table 3, Appendix A).

Postoperative hospital complications of CABG are presented in Figure 2. In the group of carbohydrate disorders, there was a higher incidence of postoperative stroke—all strokes occurred in the ICR group, the differences were significant (*p* = 0.044). In addition, The CMD group had a larger percentage of significant complications in comparison with the normoglycemic group (25.5% vs. 20.9%, respectively), but the differences were not statistically significant. The incidence of other hospital complications of CABG in the groups was comparable. A significantly larger proportion of patients in the CMD group were in the hospital after CABG > 30 days (23.4 and 13.6%, *p* = 0.014) (Figure 2).

Binary logistic regression analysis was performed to identify the factors associated with the combined endpoint (Table 4). The following factors were included in the analysis model: gender, age, BMI, diabetes mellitus, any CMD, excessive weight or obesity, Disse index, echocardiography parameters (aortic size, LV myocardial mass, LV myocardial mass index, E/ratio, Vf, e’), biochemical parameters (glucose, triglycerides), heart rate, heart failure, NYHA grade, as well as the duration of the bypass and aortic clamping time.

An increase in left atrial size, age, time of aortic clamping, and a decrease in the Disse index were associated with a significant increase in the likelihood of developing a combined endpoint. The addition of the Disse index increased the significance of the model (the Nagelkerke R-square at stage 4 was 0.743 (Appendix A).

As a second option, we targeted metabolic factors in binary logistic regression as likely predictors of the combined endpoint. The original model included gender, age, IR indices (QUICKI, Revised-QUICKI, Disse index), glucose, insulin, free fatty acids, triglycerides, body mass index. Only the Disse index, age and body mass index have shown their predictive role as predictors of long hospital stay or poor outcome (Table 5, Appendix A). Fasting glucose, insulin, lipid, QUICKI and Revised-QUICKI scores were not associated with the study outcome even at the one-way analysis stage.

## 4. Discussion

The present study shows that screening for CMD before coronary artery bypass grafting can significantly increase the number of detected carbohydrate metabolism disorders (prediabetes and type 2 diabetes). A higher rate of in-hospital complications of CABG and prolonged hospital stay was observed in the group with CMD, with the Disse insulin resistance index among the independent predictors of prolonged hospital stay or poor outcome.

The importance of active detection of latent disorders of carbohydrate metabolism before myocardial revascularization may raise doubts. On one hand, it is known that they are associated with the development of perioperative complications to the same extent as previously known diabetes mellitus [9], and are associated with a poor prognosis over long-term follow-up [24]. Therefore, identification of such patients and adequate treatment of this concomitant disease should have a beneficial effect on the prognosis. However, not all studies have shown this. For example, in a study by Djupsjo C et al. [11], patients before CABG without previously known diabetes underwent a glucose tolerance test. At 10-year follow-up after surgery, survival in the groups with normoglycemia, pre-diabetes and newly diagnosed diabetes was comparable, including after multivariate adjustment [11]. Perhaps the most optimal method for detecting CMD is the assessment of glycated hemoglobin; however, contradictory results were obtained using it as well [18,25,26,27]. An association of a high level of HbA1c with the development of perioperative complications was noted [27]; patients with diabetes showed an increased risk of death with HbA1c levels above 9.0%, as well as for the higher rate combined endpoint (death or major adverse cardiac events (MACE) with HbA1c levels above 8.1% in a five-year follow-up after CABG [26]. In addition, a meta-analysis has shown that higher preoperative HbA1c levels can potentially increase the risk of surgical site infections, renal failure, and myocardial infarction in diabetic patients after CABG, and increase the risk of death and renal failure in nondiabetic patients. However, there remains a great deal of inconsistency in the definition of high HbA1c thresholds, and there is still a need for high-quality RCTs [16]. Moreover, a study by Aydınlı B et al. [18] found no association of high HbA1c levels with immediate CABG results. The reasons for such contradictions during long-term follow-up of patients can be numerous; one of them is shown in the study by Funamizu T, et al., where strict control of diabetes achieving the HbA1c level < 6.5 in the group of diabetes patients after PCI led to a worsening prognosis during 10-year follow-up [28]. Another possible reason is the suboptimal selection of the CMD marker. Therefore, the search for additional indicators continues. For example, an increase in the level of fructosamine before CABG turned out to be associated with the development of postoperative complications [29].

Assessment of insulin resistance is another possible path. Myocardial dysfunction was noted and vascular function was impaired in patients with IR. High HOMA-IR level was associated with a decrease in global longitudinal stress and an increase in arterial stiffness [30], and HOMAIR positively correlates with the development of coronary spasm during acetylcholine provocation test [31]. An association of IR with the severity of coronary artery disease in patients with coronary artery disease has been demonstrated as well [17]. In a PROSPECT study, patients with acute coronary syndromes and IR (HOMA-IR ≥ 5) are associated with more echo-lucent plaques and increased risk of MACE compared to patients with normal IR [32]. An AIRE study on a small patient sample who underwent coronary revascularization by PCI HOMA-IR appeared to be independently associated both to de novo CAD and overall new PCI [33].

The issue of IR in the perioperative period of CABG is less studied. In a study by Aydin E et al. that investigated patients in the preoperative period, the groups with high (HOMA-IR > 2.5) and low IR did not differ in fasting glucose and HbA1c levels. When comparing the immediate results of CABG, there were no differences in the frequency of inotropes use, postoperative MI, development of rhythm disturbances, presence of infection, duration of treatment in the ICU and duration of hospital stay between groups with different IRs; however, the authors admit that modest sample size was a limitation of their study [18]. In our study, we were able to show the association of IR with immediate CABG results. Perhaps the reason for this was the use of more informative markers of IR (Disse index), assessment of the combined endpoint (complications + duration of hospitalization more than 10 days), rather than individual complications, as well as a larger number of observations. Our results are consistent with the result of Nyström T et al. [34], who carried out a long-term follow-up of patients with type 2 diabetes after CABG. They showed that a low estimated glucose disposal rate (a marker of IR) was associated with an increased risk of long-term mortality from all causes, which did not depend on other cardiovascular and metabolic risk factors. In our study, we showed the value of IR assessment in a general cohort of patients, both with the presence of CMD and with normoglycemia.

In contrast to the above studies, we used a different set of markers of insulin resistance, which are rarely used in coronary artery disease patients (FFA is the only marker that has numerous studies dedicated to it). It was previously shown that a high FFA level is an independent predictor of MACE in stable, angiographically confirmed coronary artery disease patients with different glucose metabolism statuses [35]. The Disse index is still rarely used in the assessment of insulin resistance, there are only a few publications on this issue [19,23]. Thus, Disse et al. evaluated a new lipid-parameter-based index of insulin resistance in 70 normoglycaemic non-obese individuals. The correlation coefficient between the Disse index and insulin sensitivity was higher than those with the most commonly used fasting surrogate indices for insulin sensitivity (for example, HOMA-IR, QUICKI, revised QUICKI) [23]. In another study, a strong correlation between Disse index and hyperinsulinemic–euglycemic clamp in non-diabetic post-menopausal overweight and obese women was observed, before and after weight loss intervention. This association was higher than those of HOMA, QUICKI, and McAuley indices while no significant difference was observed with Revised-QUICKI [36]. When comparing the diagnostic ability of IR indices to identify uncomplicated metabolic syndrome, the Disse index turned out to be quite informative, second only to the McAuley index. The AUC for the McAuley index was significantly greater (AUC = 0.85) than the AUCs for other indices. The second greatest AUC value was obtained using the Disse index (AUC = 0.78) [19]. In this study, we—for the first time—assessed the prognostic capabilities of the Disse index in patients with coronary artery disease. The present study showed that the determination of this particular IR marker, the Disse index, is most associated with the development of postoperative complications of CABG. Clarification of the possibility of using these IR markers in clinical practice requires further research. Additionally, IVS and LVEF turned out to be important determinants of the Disse index; accordingly, the Disse index may be associated with the results of inpatient treatment not only through metabolic but also hemodynamic factors. However, at the moment there are few data to discuss such a pathway of the Disse index effect on treatment results; further research is required to clarify this pathway.

### Study Limitation

A number of limitations should be taken into account when evaluating research results. First, this study did not compare the Disse index with other common IR indices (HOMA-IR, QUICKI, revised QUICKI, McAuley index). Although, in a previous study, it was shown that the Disse index is either superior or not inferior to them, nevertheless, the lack of a direct comparison of their predictive value is a limitation of the study. Second, OGTT was not performed in all study participants without known diabetes, but only in subjects with higher fasting glucose levels or previously known prediabetes. Third, a number of diabetic patients were treated with insulin, sulfonylurea drugs or DPP-VI inhibitors/GLP-1 analogues, which may increase serum insulin concentrations and consequently affect the levels of the computed surrogate measures of insulin sensitivity. Finally, we did not assess the level of proteinuria in the present study. It is known that the presence of mild proteinuria and diabetes presents a higher risk of mortality and cardiovascular events with long-term follow-up [37]. However, the relatively high level of glomerular filtration rate and the short follow-up period in the present study suggests that this factor was unlikely to have additional prognostic value. We also did not purposefully assess the thoroughness of glycemic control in individual patients, although the quality of glycemic control could impact the clinical outcomes [38].

## 5. Conclusions

Screening for CMD prior to coronary artery bypass grafting can significantly increase the number of patients with diagnosed disorders of carbohydrate metabolism (prediabetes and type 2 diabetes). Significant in-hospital CABG complications are more prevalent in the group with CMD. The Disse Insulin Resistance Index is an independent predictor of the combined endpoint (long hospital stay or perioperative complications).

## Figures and Tables

**Figure 1 jpm-11-00802-f001:**
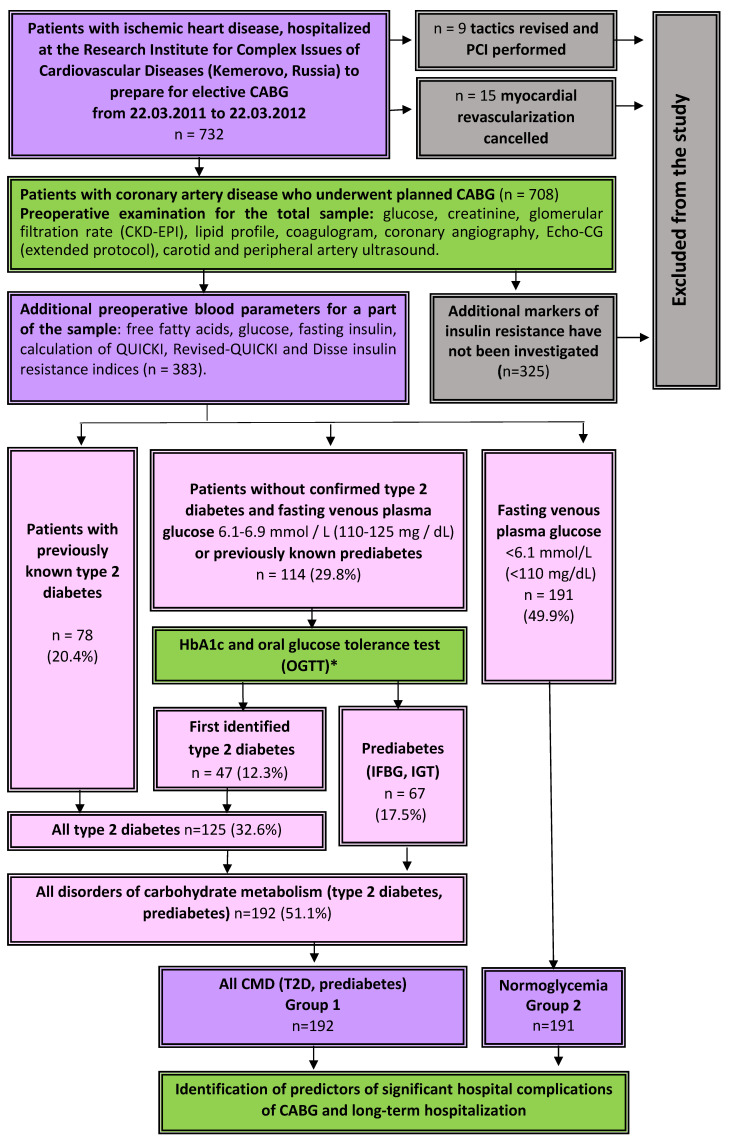
Study design. Notes: PCI—percutaneous coronary intervention, CABG—coronary artery bypass grafting, IFBG—impaired fasting blood glucose, IGT—impaired glucose tolerance, CMD—carbohydrate metabolism disorders.

**Figure 2 jpm-11-00802-f002:**
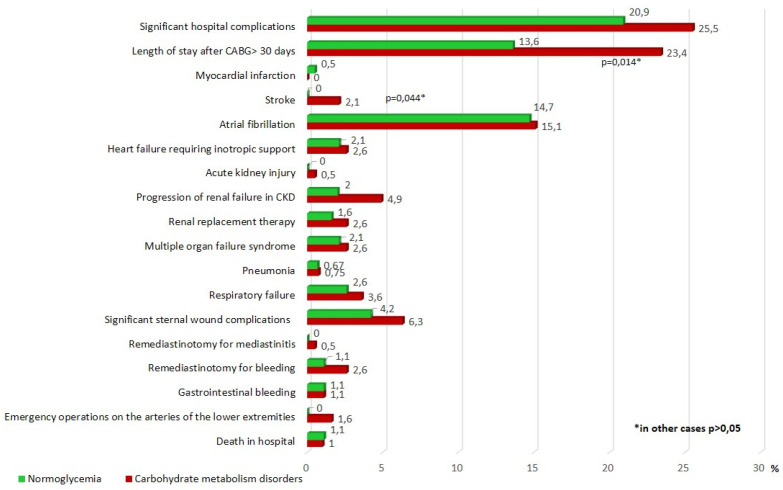
Postoperative hospital complications after CABG. Notes: CABG—coronary artery bypass grafting, CKD—chronic kidney disease, *p*—significance of differences between groups, *—*p* < 0.05.

**Table 1 jpm-11-00802-t001:** Anamnestic and clinical characteristics of patients.

	Group 1 CMD *n* = 192	Group 2 Normoglycemia *n* = 191	*p*
Men (*n*, %)	130 (67.7)	154 (80.6)	0.004
Age (years, Me [LQ; UQ])	59.0 [54.5; 63.0]	59.0 [54.0; 65.0]	0.493
Type 2 diabetes (*n*, %)	125 (65.1)	-	-
Prediabetes (IFG, IGT) (*n*, %)	67 (34.9)	-	-
BMI (kg/m^2^, Me [LQ; UQ])	29.5 [27.1; 32.5]	27.0 [24.2; 30.8]	<0.001
Obesity (BMI ≥ 30 kg/m^2^, *n*,%)	91 (47.4)	58 (30.5)	<0.001
Arterial hypertension (*n*, %)	178 (92.7)	165 (86.4)	0.043
Angina class III–IV (*n*, %)	75 (39.1)	71 (37.1)	0.701
Heart failure class NYHA III (*n*, %)	53 (27.6)	49 (25.6)	0.674
Ventricular arrhythmias (*n*, %)	28 (14.6)	26 (13.6)	0.781
Supraventricular arrhythmias (*n*, %)	20 (10.4)	13 (6.8)	0.412
Intermittent claudication (*n*, %)	24 (12.5)	29 (15.8)	0.447
Smoking (*n*, %)	49 (25.5)	79 (41.4)	<0.001
Myocardial infarction history (*n*, %)	121 (63.0)	121 (63.3)	0.947
Stroke history (*n*, %)	12 (6.2)	15 (7.8)	0.539
Previous PCI (*n*, %)	21 (10.9)	16 (8.3)	0.396
Coronary artery bypass grafting (*n*, %)	2 (1.0)	2 (1.1)	0.996
Intervention on the carotid arteries (*n*, %)	10 (5.2)	1 (0.5)	0.006
Intervention on the arteries of the lower extremities or amputation (*n*, %)	2 (1.0)	1 (0.5)	0.565
EuroSCORE II (%, Me [LQ; UQ])	1.89 [1.22; 2.89]	1.68 [1.2; 2.6]	0.166
CABG Characteristics			
Cardiopulmonary bypass (*n*, %)	177 (92.2)	169 (88.5)	0.220
Isolated coronary artery bypass grafting (*n*, %)	174 (90.6)	178 (93.2)	0.343
Combined surgery (*n*, %)	18 (9.4)	13 (6.8)	0.181
∙ Carotid endarterectomy (*n*, %)	5 (2.6)	3 (1.6)	0.479
∙ Ventriculoplasty (*n*, %)	9 (4.7)	5 (2.6)	0.280
∙ Radiofrequency ablation (*n*, %)	4 (2.1)	9 (4.7)	0.161
∙ Mitral valve (*n*, %)	0 (0)	1 (0.5)	0.315
∙ Aortic valve (*n*, %)	1 (0.5)	2 (1.1)	0.559
Cardiopulmonary bypass duration (minutes, Me [LQ; UQ])	98.0 [79.0; 116.0]	95.0 [78.0; 109.0]	0.229
Aortic clamping time (minutes, Me [LQ; UQ])	61.5 [50.0; 75.0]	60 [49.0; 72.0]	0.331
Total duration of surgery (minutes, Me [LQ; UQ])	246 [204.0; 298.0]	240.0 [198.0; 264.0]	0.152
Intraoperative blood loss (ml, Me [LQ; UQ])	500.0 [500.0; 600.0]	500.0 [500.0; 550.0]	0.241
Number of shunts (Me [LQ; UQ ])	3.0 [2.0; 3.0]	2.0 [2.0; 3.0]	0.352
Complete revascularization (*n*, %)	178 (92.7)	172 (90.1)	0.354
LOS after CABG (days, Me [LQ;UQ])	13.0 [11.0; 16.0]	12.0 [10.0; 14.0]	0.003
LOS after CABG > 10 days (*n*, %)	151 (78.6)	126 (66.3)	0.007
LOS after CABG > 30 days (*n*, %)	45 (23.4)	26 (13.7)	0.014
Preoperative Drug Therapy			
β-blockers (*n*, %)	190 (98.9)	186 (97.3)	0.241
Angiotensin-converting enzyme inhibitors (*n*,%)	159 (82.8)	158 (82.7)	0.956
Angiotensin 2 receptor antagonists (*n*,%)	11 (5.7)	5 (2.6)	0.300
Mineralocorticoid receptor antagonists (*n*, %)	34 (17.7)	32 (16.8)	0.765
Thiazide-like diuretics (*n*, %)	19 (9.8)	18 (9.4)	0.916
Loop diuretics (*n*, %)	139 (72.4)	111 (58.2)	0.003
Calcium channel blockers (*n*, %)	182 (68.4)	254 (57.4)	0.007
Only oral antihyperglycemic drugs	41 (21.3)	-	-
Metformin	72 (37.6)	-	-
Sulfonylurea drugs	38 (19.7)	-	-
Inhibitors DPP 4/GLP 1 receptor agonists	5 (2.6)	-	-
Insulin therapy before hospitalization	19 (9.9)	-	-
Insulin therapy during hospitalization	55 (28.6)	-	-

Notes: CMD—carbohydrate metabolism disorders, IFG—impaired fasting glucose, IGT-impaired glucose tolerance, BMI—body mass index, FC—functional class, NYHA—New York Heart Association, PCI—percutaneous coronary intervention, LOS—length of stay, CABG—coronary artery bypass grafting, EuroSCORE II—European System for Cardiac Operative Risk Evaluation, DPP 4—dipeptidyl peptidase 4, GLP 1—lucagon-like peptide 1.

**Table 2 jpm-11-00802-t002:** Preoperative laboratory and instrumental parameters.

	Group 1 CMD *n* = 192	Group 2 Normoglycemia *n* = 191	*p* Value
Preoperative Fasting Blood Laboratory Parameters (Me [LQ; UQ])			
Total cholesterol (mmol/L)	5.0 [4.0; 6.1]	5.0 [4.2; 6.0]	0.989
HDL cholesterol (mmol/L)	0.9 [0.8; 1.1]	1.0 [0.9; 1.2]	0.031
LDL cholesterol (mmol/L)	2.9 [2.2; 3.8]	2.9 [2.3; 3.7]	0.654
Triglycerides (mmol/L)	2.0 [1.4; 2.5]	1.6 [1.2; 2.2]	<0.001
Creatinine (μmol/L)	82.0 [69.0; 98.5]	83.0 [74.0; 106.0]	0.158
GFR CKD-EPI (ml/min/1.73 m^2^)	82.0 [66.5; 99.7]	82.4 [66.3; 103.5]	0.190
Prothrombin index	100.0 [89.0; 108.0]	100.0 [89.0; 108.0]	0.894
APTT (sec)	30.0 [26.3; 35.9]	30.0 [27.5; 36.0]	0.200
Thrombin time (seconds)	15.6 [14.8; 16.3]	15.6 [14.8; 16.3]	0.795
Fibrinogen (g/L)	4.4 [3.7; 6.0]	4.4 [3.5; 5.6]	0.286
SFMS (g/L)	5.5 [4.0; 10.0]	5.0 [4.0; 8.0]	0.218
Glycated hemoglobin (HbA1c, %)	7.3 [6.9; 8.0]	5.0 [4.8; 5.6]	<0.001
Glucose, venous plasma (mmol/L)	6.7 [6.1; 8.2]	5.2 [4.9; 5.5]	<0.001
Insulin, IU/mL	7.64 [1.83; 24.30]	10.1 [2.98; 20.84]	0.729
Free fatty acids, mmol/L	0.41 [0.28; 8.8]	0.33 [0.22; 0.48]	<0.001
Disse index	−13.03 [−26.33; −3.88]	−12.48 [−21.68; −7.16]	0.811
QUICKI	0.151 [0.126; 0.184]	0.146 [0.134; 0.178]	0.414
Revised-QUICKI	0.174 [0.145; 0.222]	0.172 [0.154; 0.266]	0.367
Preoperative Echocardiogram (Me [LQ; UQ])			
LV end-diastolic volume (mL)	160.0 [136.0;194.0]	154.0 [132.5; 185.0]	0.042
LV end-diastolic dimension (cm)	5.6 [5.3; 6.2]	5.5 [5.1; 6.0]	0.135
LV end-systolic volume (mL)	66.0 [51.0; 101.0]	59.5 [44.0; 91.0]	0.009
LV end-systolic dimension (cm)	3.9 [3.5; 4.7]	3.7 [3.3; 4.5]	0.006
Interventricular septum (cm)	1.1 [1.0; 1.2]	1.1 [1.0–1.2]	0.808
Posterior wall of the LV (cm)	1.1 [0.9; 1.2]	1.1 [1.0–1.2]	0.451
Left atrium (cm)	4.3 [4.0; 4.5]	4.2 [3.8; 4.4]	0.009
LV aneurysm (*n*, %)	4 (2.1)	1 (0.5)	0.408
LV ejection fraction (%)	59.0 [50.0; 64.0]	62.0 [52.0; 65.0]	0.037
Mean pulmonary artery pressure (mmHg)	15.0 [12.0; 27.0]	18.0 [12.0; 28.0]	0.782
E/A—ratio of early and late diastolic transmitral flow	0.8 [0.7; 1.1]	0.8 [0.7; 1.2]	0.189
Vf—Flow propagation velocity (cm/sec)	46.5 [40.0 60.0]	48.0 [45.0; 60.0]	0.155
LV myocardial mass by Deveraux and Reichek (g)	312.0 [258.5; 372.0]	292.5 [241.1; 370.0]	0.029
LV myocardial mass index (g/m^2^)	135.8 [160.5; 188.0]	155.0 [126.2; 188.1]	0.119
Stroke volume (ml)	90.5 [81.0; 99.5]	89.0 [76.0; 103.0]	0.113
LV—relative wall thickness index	0.4 [0.3; 0.4]	0.4 [0.3; 0.4]	0.628
E/Vf	1.3 [1.0; 1.6]	1.2 [1.1; 1.5]	0.633
Data of Instrumental Examinations of the Coronary and Non-Coronary Arteries			
1-vessel disease *	56 (21.1)	101 (22.9)	0.585
2-vessel disease *	69 (25.9)	121 (27.4)	0.495
3-vessel disease *	121 (45.5)	189 (42.8)	0.521
Left Main Coronary Artery Stenosis > 50%	54 (20.3)	77 (17.4)	0.282
Average thickness of the intima-media complex (mm, Me [LQ; UQ])	1.2 [1.0; 1.2]	1.1 [1.0; 1.2]	0.246
Hemodynamically significant stenosis of the carotid arteries (50% or more *n*, %)	64 (24.1)	93 (21.0)	0.401
Hemodynamically significant stenosis of the arteries of the lower extremities (*n*, %)	80 (30.1)	152 (34.4)	0.249

Notes: CMD—carbohydrate metabolism disorders, Me [LQ; UQ]—median with upper and lower quartile, HDL—high-density lipoprotein, LDL—low-density lipoprotein, GFR—glomerular filtration rate, CKD-EPI—Chronic Kidney Disease Epidemiology Collaboration, APTT—complex activated partial thromboplastin time, SFMS—soluble fibrin monomer complexes, QUICKI—quantitative insulin sensitivity check index, LV—left ventricular, E—the rate of early diastolic filling of the LV, A—late diastolic filling rate of the LV, E/A—ratio of early and late diastolic transmitral flow, IVRT—isovolumic relaxation time LV, E/Vf—peak ratio of the early transmitral flow to the propagation velocity of the early diastolic flow, *—the number of involved main coronary arteries.

**Table 3 jpm-11-00802-t003:** Multiple linear regression to assess the relationship between FFA and Disse index with patients’ perioperative characteristics.

Model		Unstandardized Coefficients		Standardized Coefficients	t	*p* Value Sig.
		B	Std. Error	Beta		
Free Fatty Acids ^a^						
1	(Constant)	0.210	0.046		4.592	0.000
	TG	0.113	0.020	0.374	5.558	0.000
2	(Constant)	−0.097	0.108		−0.899	0.370
	TG	0.102	0.020	0.337	5.044	0.000
	HeartRate	0.005	0.002	0.209	3.120	0.002
3	(Constant)	−0.290	0.127		−2.292	0.023
	TG	0.109	0.020	0.359	5.432	0.000
	HeartRate	0.006	0.002	0.221	3.356	0.001
	APTT	0.005	0.002	0.183	2.803	0.006
4	(Constant)	−0.607	0.172		−3.542	0.001
	TG	0.095	0.020	0.313	4.655	0.000
	HeartRate	0.005	0.002	0.216	3.333	0.001
	APTT	0.005	0.002	0.187	2.904	0.004
	BMI	0.012	0.005	0.178	2.691	0.008
5	(Constant)	−0.706	0.175		−4.035	0.000
	TG	0.089	0.020	0.294	4.384	0.000
	HeartRate	0.005	0.002	0.207	3.225	0.001
	APTT	0.005	0.002	0.190	2.983	0.003
	BMI	0.011	0.005	0.156	2.358	0.019
	Glucose	0.026	0.011	0.148	2.290	0.023
Disse Index ^b^						
1	(Constant)	5.663	8.481		0.668	0.507
	IVS	−18.186	7.952	−0.311	−2.287	0.027
2	(Constant)	−6.560	9.890		−0.663	0.510
	IVS	−19.286	7.676	−0.329	−2.512	0.015
	LVEF	0.251	0.114	0.287	2.193	0.033

^a^ Dependent Variable: Free fatty acids, ^b^ Dependent Variable: Disse. TG—triglycerides. APTT—complex activated partial thromboplastin time. BMI—body mass index. IVS interventricular septum. LV left ventricular ejection fraction.

**Table 4 jpm-11-00802-t004:** Predictors of the combined endpoint (postoperative complications or length of stay > 10 days) in binary logistic regression.

		B	S.E.	Wald	df	Sig. *p* Value	Exp (B)
Step 1 ^a^	Left atrium	3.589	1.121	10.251	1	0.001	36.197
	Constant	−13.669	4.360	9.831	1	0.002	0.000
Step 2 ^b^	Age	0.144	0.070	4.189	1	0.041	1.155
	Left atrium	3.276	1.131	8.390	1	0.004	26.457
	Constant	−20.599	6.045	11.614	1	0.001	0.000
Step 3 ^c^	Age	0.197	0.087	5.089	1	0.024	1.217
	Left atrium	4.339	1.462	8.813	1	0.003	76.659
	AoClampTime	−0.057	0.028	4.051	1	0.044	0.945
	Constant	−24.483	7.410	10.916	1	0.001	0.000
Step 4 ^d^	Disse index	−0.144	0.078	3.460	1	0.063	0.865
	Age	0.276	0.118	5.435	1	0.020	1.318
	Left atrium	4.429	0.743	6.453	1	0.011	83.828
	AoClampTime	−0.073	0.034	4.499	1	0.034	0.930
	Constant	−30.057	0.773	9.459	1	0.002	0.000

^a^. Variable (s) entered on step 1: LA, ^b^. Variable (s) entered on step 2: Age, ^c^. Variable (s) entered on step 3: AoClampTime, ^d^. Variable (s) entered on step 4: Disse, LA—left atrial size, AoClampTime—time of aortic clamping, Disse—index Disse.

**Table 5 jpm-11-00802-t005:** Metabolic predictors of the combined endpoint (poor outcome or length of stay > 10 days) in binary logistic regression.

Variables in the Equation							
		B	S.E.	Wald	df	Sig.	Exp (B)
Step 1 ^a^	Age	0.091	0.027	11.182	1	0.001	1.096
	Constant	−4.097	1.553	6.963	1	0.008	0.017
Step 2 ^b^	Disse	−0.058	0.020	8.749	1	0.003	0.943
	Age	0.115	0.030	14.621	1	0.000	1.122
	Constant	−6.241	1.797	12.060	1	0.001	0.002
Step 3 ^c^	Disse	−0.060	0.021	8.204	1	0.004	0.942
	Age	0.121	0.031	15.133	1	0.000	1.129
	BMI	0.124	0.048	6.632	1	0.010	1.132
	Constant	−10.156	2.460	17.047	1	0.000	0.000

^a^. Variable (s) entered on step 1: Age, ^b^. Variable (s) entered on step 2: Disse, ^c^. Variable (s) entered on step 3: BMI. BMI—body mass index, Disse—Disse index.

## Data Availability

Data sharing not applicable.

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
