# Peer review of "Screening for Glucose Metabolism Disorders, Assessment the Disse Insulin Resistance Index and Hospital Prognosis of Coronary Artery Bypass Surgery"

_jpm, 2021, doi:10.3390/jpm11080802_

Round 1

Reviewer 1 Report

In this manuscript, the authors investigated the role of insulin resistance markers in the post-operative complications of coronary artery bypass grafting (CABG) in 383 patients They found that patients with DM and IGT had more frequent hospital complications and the Disse Index was an independent predictor of long hospital stay and/or poor outcome. It was well written and the findings and conclusion could be well appreciated, however, there are several points the authors should address.

Major

In the first half part, the authors compared the characteristics and outcomes between the CMD and Normoglycemia. In the second half, they focus on the Disse index as a predictor of combined outcomes using several multivariate analysis although there is no difference in Disse index between CMD and normoglycemia group. It will be helpful to demonstrate the patients characteristics among the high or low Disse index. In the endpoint analysis, the authors used combined endpoint of a hospital stay > 10 days, and other in-hospital complication. Usually the hospital stay would be longer if the patients had an in-hospital complication. What was the reason of longer hospital stay other than major complication?

Minor

  • In the several multivariate analysis, the authors should describe how and why did the authors select the variables.
  • The discussion of the relation between the FFA and Disse index appears to be beyond the scope oof this manuscript.
  • The authors indicated IVUS and LVEF as significant determinants of Disse index that suggests the Disse index may be related the in-hospital outcomes through not only the metabolic but also hemodynamic factors. Further discussion is preferable.

Author Response

Dear reviewer, thank you for the work done and the comment made!

Major:

In the first half part, the authors compared the characteristics and outcomes between the CMD and Normoglycemia. In the second half, they focus on the Disse index as a predictor of combined outcomes using several multivariate analysis although there is no difference in Disse index between CMD and normoglycemia group.

Indeed, we were also somewhat surprised by the absence of differences in the value of the Disse index between the groups with CMD and normoglycemia. Our assumption is that among patients with severe coronary atherosclerosis and in the group without established CMD, insulin resistance may occur, but these violations do not meet the CMD criteria.

It will be helpful to demonstrate the patients characteristics among the high or low Disse index.

There are no reference values for many calculated indices of insulin resistance, including the Disse index, and therefore it is impossible to divide the sample into a group with high and low Disse index.

  There is only a general tendency, a regularity - an increase in Disse index indicates a decrease in insulin resistance (despite the fact that the Disse index itself always has a negative value)

In the endpoint analysis, the authors used combined endpoint of a hospital stay > 10 days, and other in-hospital complication. Usually the hospital stay would be longer if the patients had an in-hospital complication. What was the reason of longer hospital stay other than major complication?

Thank you for your comment. In this article, we took into account only significant complications; at the same time, complications that are not significant, but nevertheless require a hospital stay (for example, minimal hydrothorax, pneumothorax), could increase the time of hospitalization after surgery. The combined endpoint was a obligatory measure in order to increase the number of outcomes taken into account in statistical calculations.

Minor:

  • In the several multivariate analysis, the authors should describe how and why did the authors select the variables.

Dear reviewer, this information is available in the article (see below). We have posted them in the research results section.

First, A multiple regression was run to predict FAA from age, body mass index, Eu-roSCORE II, aortic clamping time, bypass duration, APTT, SFMS, fibrinogen, glucose, triglycerides, LV myocardial mass index, length of stay, heart rate, left atrium. When predicting the relationship of factors with the Disse index, age, body mass index, EuroSCORE II, heart rate, aortic clamping time, bypass duration, length of stay, echocardiographic data, free fatty acids, triglycerides, glucose, APTT, fibrinogen, SFMS were included in the multiple regression model.

Second, binary logistic regression analysis was performed to identify the factors associated with the combined endpoint (Table 4). The following factors were included in the anal-ysis model: gender, age, BMI, diabetes mellitus, any CMD, excessive weight or obesity, Disse index, echocardiography parameters (aortic size, LV myocardial mass, LV myo-cardial mass index, E / ratio, Vf, e'), biochemical parameters (glucose, triglycerides), heart rate, heart failure, NYHA grade, as well as duration of the bypass and aortic clamping time.

Third, we targeted metabolic factors in binary logistic regression as likely predictors of the combined endpoint. The original model included gender, age, IR indices (QUICKI, Revised-QUICKI, Disse index), glucose, insulin, free fatty acids, triglycerides, body mass index. Only the Disse index, age and body mass index have shown their predictive role as predictors of long hospital stay or poor outcome

  • The discussion of the relation between the FFA and Disse index appears to be beyond the scope oof this manuscript.

Indeed, our article did not discuss the relationship between FFA and the Disse index. First of all, we tried to identify the patterns in which this index manifests itself. In future research, we will analyze this association more thoroughly.

  • The authors indicated IVUS and LVEF as significant determinants of Disse index that suggests the Disse index may be related the in-hospital outcomes through not only the metabolic but also hemodynamic factors. Further discussion is preferable.

Thanks for this comment. Indeed, IVS and LVEF turned out to be important determinants of the Disse index; accordingly, the Disse index may be associated with the results of inpatient treatment not only through metabolic, but also hemodynamic factors. However, at the moment there is little data to discuss such pathway of the Disse index effect on treatment results; further research is required to clarify this pathway.

Reviewer 2 Report

A rather well conducted registry study on the effect of diabetes mellitus and its precursors on outcome in CABG-patients.

The present study analyses, whether screening for parameters of glucose metabolism might provide prognostic information on the clinical course of patients undergoing coronary artery bypass operation (CABG) This is a relevant topic, as one might probably have a more thorough view on these patients after the operation. The present study is the first addressing this question in patients scheduled for CABG and the authors discuss their results extensively with other studys on patients with coronary artey disease. The paper is well written and the conclusions are consistent with the results.

Author Response

Dear Reviewer! Thank you very much for the work done in evaluating our article and high opinion of the manuscript.

Round 2

Reviewer 1 Report

No further comments.